# Central retinal vascular trunk deviation in unilateral normal-tension glaucoma

Ho-Kyung Choung[1,2], Martha Kim[3], Sohee Oh [4], Kyoung Min Lee [1,2]*, Seok Hwan Kim[1,2]

1 Department of Ophthalmology, Seoul National University College of Medicine, Seoul, Korea, 2 Department of Ophthalmology, Seoul National University Boramae Medical Center, Seoul, Korea, 3 Department of Ophthalmology, Dongguk University Ilsan Hospital, Goyang, Korea, 4 Department of Biostatistics, Seoul National University Boramae Medical Center, Seoul, Korea

* isletzz@gmail.com

## Abstract

### Purpose

To investigate whether the position of the central retinal vascular trunk (CRVT), as a surrogate of lamina cribrosa (LC) offset, was associated with the presence of glaucoma in normal-tension glaucoma (NTG) patients.

### Methods

The position of the CRVT was measured as the deviation from the center of the Bruch's membrane opening (BMO), as delineated by spectral-domain optical coherence tomography imaging. The offset index was calculated as the distance of the CRVT from the BMO center relative to that of the BMO margin. The angular deviation of CRVT was measured with the horizontal nasal midline as 0˚ and the superior location as a positive value. The offset index and angular deviation were compared between glaucoma and fellow control eyes within individuals.

### Results

NTG eyes had higher baseline intraocular pressure ($P$ = 0.001), a larger β-zone parapapillary atrophy area ($P$ = 0.013), and a larger offset index ($P$<0.001). In a generalized linear mixed-effects model, larger offset index was the only risk factor of NTG diagnosis (OR = 31.625, $P$<0.001). A generalized estimating equation regression model revealed that the offset index was larger in the NTG eyes than in the control eyes for all ranges of axial length, while it was the smallest for the axial length of 23.4 mm (all $P$<0.001).

### Conclusions

The offset index was larger in the unilateral NTG eyes, which fact is suggestive of the potential role of LC/BMO offset as a loco-regional susceptibility factor.

**Data Availability Statement:** All relevant data are within the manuscript.

**Funding:** This work was supported by a clinical research grant-in-aid from the Seoul National University Boramae Medical Center (grant no. 04-

2021-0011) (KML). The funders had no role in study design, data collection and analysis, decision to publish, or preparation of the manuscript.

**Competing interests:** The authors have declared that no competing interests exist.

## Introduction

Glaucoma is characterized by progressive axonal loss of retinal ganglion cells [1–3]. Many factors such as mechanical, ischemic, metabolic, and immunologic insults have been nominated as candidate sources of axonal damage [1–3]. So far, intraocular pressure (IOP) is the only controllable factor, but it cannot explain every aspect of glaucoma. To be specific, in many cases, glaucomatous damage starts with a localized retinal nerve fiber layer (RNFL) defect, whereas IOP affects the optic nerve head (ONH) universally. This suggests a loco-regional susceptibility factor that makes some parts of the ONH more vulnerable to damage.

The lamina cribrosa (LC) is the principal site of axonal injury in glaucoma [4]. Therefore, LC change might be involved in increased loco-regional susceptibility. In the recent Boramae Myopia Cohort Study, we found that axial elongation evoked LC shift, in contrast to the relative preservation of the Bruch's membrane opening (BMO) [5–7]. In subsequent studies, the direction of LC offset from the BMO center showed a strong spatial correlation with the initial hemispheric location of glaucomatous damage in both myopic normal-tension glaucoma (NTG) [8] and myopic high-tension glaucoma [9]. This indicated that LC/BMO offset might reflect the loco-regional susceptibility of the ONH.

Besides the initial asymmetric hemispheric involvement of glaucomatous damage, unilateral involvement of glaucoma is often encountered. Even if fellow eyes are not normal and might, therefore, be affected by glaucoma in the future, unilateral glaucoma eyes are the best subjects to be scrutinized for loco-regional susceptibility factors after adjusting for systemic factors within a subject at the time of study. The purpose of the present study was to determine whether the position of the central retinal vascular trunk (CRVT), as a surrogate of LC offset, increases susceptibility to glaucomatous damage in unilateral NTG patients while other systemic risk factors are controlled by inter-eye comparison within the same subject.

## Methods

This investigation was based on NTG patients included in the Boramae Glaucoma Imaging Study (BGIS), an ongoing prospective study at Seoul National University Boramae Medical Center (Seoul, Korea). Written informed consent to participate was obtained from all of the subjects. The study protocol was approved by the Seoul National University Boramae Medical Center Institutional Review Board and conformed to the tenets of the Declaration of Helsinki.

All of the participants underwent a full ophthalmologic examination that included best-corrected visual acuity (BCVA) assessment, refraction, slit-lamp biomicroscopy, Goldmann applanation tonometry, gonioscopy, dilated funduscopic examination, keratometry (RKT-7700; Nidek, Hiroshi, Japan), axial length measurement (IOLMaster version 5; Carl Zeiss Meditec, Dublin, CA, USA), disc photography and red-free fundus photography (TRC-NW8; Topcon, Tokyo, Japan), spectral-domain optical coherence tomography (SD-OCT; Spectralis OCT, Heidelberg Engineering, Heidelberg, Germany) and standard automated perimetry (Humphrey Field Analyzer II 750, 24–2 Swedish Interactive Threshold Algorithm; Carl-Zeiss Meditec, Dublin, CA, USA). During the acquisition of SD-OCT images, the subjects were asked to fixate on the target, and images were acquired with the forehead and chin stabilized by the headrest. Extra care was taken during each exam to confirm that the forehead and chin were correctly positioned and did not move. Prior to treatment, IOP was measured repeatedly (typically five times) on the same or different days. The average value, which was defined as the baseline IOP, was used for the subsequent analysis.

Glaucomatous optic nerve damage was defined as rim thinning, notching and the presence of RNFL defects, and was evaluated by a glaucoma specialist (SHK). NTG was defined as glaucomatous optic nerve damage and associated visual field defects, an open iridocorneal angle

(in the case of cataract surgery, the angle was confirmed preoperatively), and IOP $\leq$ 21mmHg at any point before or after treatment. Glaucomatous visual field defect was defined as (1) outside normal limits on glaucoma hemifield test, or (2) three abnormal points, with a *P* value less than 5% probability of being normal and one with a *P* value less than 1% by pattern deviation, or (3) pattern standard deviation of less than 5%. Visual field defects were confirmed on two consecutive reliable tests (fixation loss rate of $\leq$ 20%, and false-positive and false-negative error rates of $\leq$ 25%).

The inclusion criteria were unilateral NTG patients. To be specific, the contralateral eye for each patient had to have a normal optic disc appearance, an open iridocorneal angle, a normal red-free fundus photograph, and a normal visual field. The exclusion criteria were BCVA of < 20/40, a sharply defined posterior staphyloma (which can deform the contour of the eyeball) on funduscopic examination, a history of ocular surgery other than cataract extraction or corneal refractive surgery, retinal or neurologic disease other than glaucoma that could cause visual field defect, a poor-quality image (i.e., quality score <15) of any section on enhanced depth imaging (EDI) SD-OCT radial scans, and a CRVT position located within the BMO but impossible to determine clearly due to vessel bifurcation. NTG eyes were compared with their fellow control eyes.

## Assessment of the central retinal vascular trunk position

Our strategy for demarcation of BMO and CRVT position has been described previously [8–10]. The peripapillary area was imaged by SD-OCT (Spectralis, Heidelberg Engineering) using EDI technique. Possibility of the magnification error was prevented by entering the corneal curvature of each eye into the SD-OCT system before scanning. The deep ONH complex was imaged using the EDI technique. The BMO was demarcated using the Glaucoma Module Premium Edition of the Spectralis machine. With 24 high-resolution radial scan images of the ONH, 15˚ apart from each other, each averaged from 24 individual B-scans, SD-OCT automatically detects the margin of the BMO. Every detected BMO margin was reviewed by one of the authors (KML), and errors were corrected manually. Based on the edited BMO margin, the Spectralis machine calculated the area and center of the BMO.

The CRVT position was determined in the same way of our previous studies [8–10]. First, the location of the CRVT was demarcated on fundoscopic infrared images and color-disc photography (Fig 1). Then, its location was confirmed by cross-sectional SD-OCT imaging in all cases. If a CRVT was not visible on either infrared fundus photographs or B-scan EDI SD-OCT images, fluorescein or OCT angiography (Spectralis) was used to determine the presence of the CRVT within the BMO. The position of the CRVT was defined in two aspects: 1) its angular deviation (Fig 1B, α), and 2) the extent of offset (Fig 1B, *a*). The angle was measured based on the right-eye orientation, with the nasal horizontal midline as 0˚ (a positive value indicating a vascular trunk located superiorly, and a negative value indicating a vascular trunk located inferiorly). To evaluate the extent of offset, the distance of the CRVT from the center of the BMO (*a*) was divided by the distance of the BMO margin from the center of the BMO in that direction (*b*), and defined as 'offset index' (Fig 1B, *a/b*). In cases of invisible CRVT due to being located outside the BMO, the offset index was defined as 1.0, and the angular deviations were not determined. Using the Image J program (version 1.51, National Institutes of Health, Bethesda, MD, USA), one of the authors (KML), who as blinded to the participants' clinical information, measured the distances and angles. The reproducibility of the locating of the CRVT was excellent, as we had stated in the previous study [8].

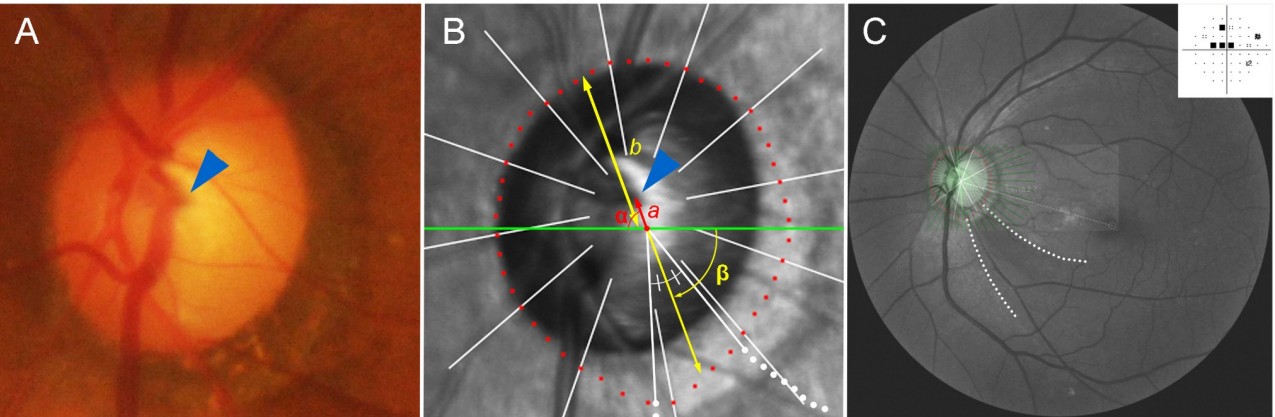

**Fig 1. Measurement of central retinal vascular trunk (CRVT) deviation.** (**A**) Disc photograph. The arrowhead indicates the emergence of the CRVT. (**B**) Infrared image obtained by spectral-domain optical coherence tomography (SD-OCT) with demarcated margin of Bruch's membrane opening (BMO). The red dots indicate the BMO margin, and the green line indicates the reference line. The angular deviation of the CRVT (α) is measured clock-wise, with the nasal horizontal midline as 0˚. A positive value indicates the superior location, and a negative value indicates the inferior location relative to the reference line. The angular location of retinal nerve fiber layer (RNFL) defect (β) is measured clock-wise, with the temporal horizontal midline as 0˚. A positive value indicates the superior location and a negative value indicates the inferior location relative to the BMO center, the distances are measured to the CRVT (*a*) and to the BMO margin in the same direction (*b*). The ratio of these distances was defined as the 'offset index' (*a/b*), which was used to measure the extent of offset. (**C**) Red-free fundus photo. A singular RNFL defect is observed in the inferior hemisphere (between the dotted lines). An infrared SD-OCT image is transposed to show the BMO margin. The Humphrey visual field result shows superior scotomas corresponding to the inferior RNFL defect.

## Assessment of RNFL defects

The angular location of RNFL defect was defined in the same image used for the CRVT localization based on the right-eye orientation [8]. First, an infrared fundus image was overlapped with a red-free fundus photograph using commercial software (Photoshop; Adobe, San Jose, CA, USA) (Fig 1C). The points where the boundaries of the RNFL defect meet the BMO margin were determined. The angular location of the midpoint of the RNFL defect on the BMO margin was measured from the center of the BMO, with the temporal horizontal midline as 0˚ (a positive value indicating an RNFL defect located superiorly, and a negative value indicating an RNFL defect located inferiorly) (Fig 1B, **β**). In cases of RNFL defect in both hemispheres, the angular location was measured to the larger RNFL defect based on the angular width.

## Assessment of ONH structures

β-zone parapapillary atrophy (PPA) was defined as the area without any retinal pigment epithelium (RPE) adjacent to the ONH, and γ-zone PPA as the area without BM within the β-zone PPA. The β-zone and γ-zone PPA areas were measured on the same infrared OCT fundus images by one observer (KML) blinded to the subjects' information. Using the built-in caliper tool of the Spectralis OCT system, two boundaries were drawn: 1) the RPE opening (RPEO), which is the area without the RPE, and 2) the clinical disc margin (CDM). The β-zone PPA area was defined as the RPEO area minus the CDM area, while the γ-zone PPA was defined as the BMO area minus the CDM area [7]. Based on deep ONH scanning using the Glaucoma Module Premium Edition of the Spectralis machine, the BMO minimum rim width (BMO-MRW) and circumpapillary RNFL thickness of 3.5mm diameter from the BMO center were also measured [11].

## Data analysis

The paired *t*-test was used for the intra-individual comparison between the NTG and fellow control eyes. Subgroup analyses were performed by the Mann-Whitney U test and Wilcoxon

signed-rank test. To account for the paired-eye correlation, a generalized linear mixed-effects model (GLMM) was used for univariable and multivariable analyses. Parameters with a *P* value less than 0.20 in the univariable analysis were included in the subsequent multivariable analysis. A locally weighted scatterplot smoothing (LOESS) curve was fitted to the data. The LOESS curve uses iterative weighted least squares to determine values that best fit the data [12]. A generalized estimating equation (GEE) regression model was applied to simulate the offset index change according to the glaucoma diagnosis and axial length while controlling for paired-eye correlations. Statistical analyses were performed with commercially available software (Stata version 14.0; StataCorp, College Station, TX, USA) and R statistical packages version 3.4.3 (available at http://www.r-project.org; assessed December 5, 2017). The data herein are presented as the mean ± standard deviation except where stated otherwise, and the cutoff for statistical significance was set at $P < 0.05$.

## Results

This study initially involved 112 unilateral NTG patients. Of these, 7 patients were excluded due to poor image quality of radial scans leading to incomplete visualization of the BMO margin, and 6 patients were excluded owing to bifurcation of CRVT on emergence. The emergence of the CRVT was not visible in 17 patients on infrared imaging. Among these, 14 patients were proved to not have the CRVT within the BMO by fluorescein or OCT angiography and they were included, while 3 who had not undergone angiography were excluded. Three patients who had the CRVT outside the BMO in both eyes were also excluded, leaving a final sample of 93 unilateral NTG patients. The subjects were aged 53.8 ± 14.4 years, 44 of whom were female (47%).

As compared with the respective fellow control eyes, NTG eyes had higher baseline IOP (14.2 ± 2.5 mmHg vs. 13.8 ± 2.3 mmHg, *P* = 0.001), longer axial length (25.2 ± 1.6 mm vs. 25.1 ± 1.5 mm, *P*<0.001), thinner average circumpapillary RNFL thickness (85.2 ± 13.8 *μm* vs. 98.5 ± 12.8 *μm*, *P*<0.001), narrower average BMO-MRW (220.5 ± 42.6 *μm* vs. 246.6 ± 47.2 *μm*, *P*<0.001), and worse visual field parameters (mean deviation: -4.20 ± 3.12 dB vs. -1.06 ± 1.48 dB, *P*<0.001; pattern standard deviation: 5.45 ± 3.68 dB vs. 1.97 ± 0.84 dB, *P*<0.001) (Table 1). The area of the BMO (2.76 ± 0.72 mm$^2$ vs. 2.65 ± 0.67 mm$^2$, *P* = 0.042), that of β-zone PPA (1.29 ± 0.83 mm$^2$ vs. 1.12 ± 0.84 mm$^2$, *P* = 0.013), and that of γ-zone PPA (0.62 ± 0.62 mm$^2$ vs. 0.46 ± 0.54 mm$^2$, *P*<0.001) were larger in the NTG eyes than in the fellow control eyes. The offset index also was larger in the NTG eyes (0.57 ± 0.27 vs. 0.39 ± 0.23, *P*<0.001), while the angular deviation of CRVT did not differ between the groups (15.2 ± 49.1˚ vs. 5.4 ± 57.4˚, *P* = 0.171). The GLMM analysis revealed that larger offset index was the only risk factor for NTG diagnosis (OR = 31.625, *P*<0.001) (Table 2).

The offset index varied according to the diagnosis and axial length (Fig 2A). The NTG group had a larger offset index than the fellow control group for all ranges of axial length. Regarding the effect of axial length, the offset index showed a non-linear relationship with the lowest value at the axial length of 23.4 mm. From this point, the offset index increased in either direction of axial length change (Fig 2A, red dashed vertical line). To evaluate the effect of both factors while accounting for the paired-eye correlations, the GEE regression model was fitted to the data set (Table 3). The equation was constructed as first-, second-, and third-order terms of the axial length and first-order term of the diagnosis. It corresponded to J-shaped curves according to axial length and showed an independent effect of diagnosis on the offset index (Table 3).

The NTG eyes had a larger offset index than did the fellow eyes in cases either of nasally located CRVT (Fig 3) or temporally located CRVT (Fig 4). To compare the intra-individual

**Table 1. Demographic comparison between glaucoma and fellow control eyes.**

|  | Glaucoma eye (N = 93) | Fellow control eye (N = 93) | $P^*$ |
|---|---|---|---|
| Baseline IOP, *mmHg* | 14.2±2.5 | 13.8±2.3 | 0.001 |
| Axial length, *mm* | 25.2±1.6 | 25.1±1.5 | 0.002 |
| Angular deviation of vascular trunk,° | 15.2±49.1 | 5.4±57.4 | 0.171 |
| Offset index | 0.57±0.27 | 0.39±0.23 | <0.001 |
| BMO area, $mm^2$ | 2.76±0.72 | 2.65±0.67 | 0.042 |
| β-zone PPA area, $mm^2$ | 1.29±0.83 | 1.12±0.84 | 0.013 |
| γ-zone PPA area, $mm^2$ | 0.62±0.62 | 0.46±0.54 | <0.001 |
| Average circumpapillary RNFL thickness, *μm* | 85.2±13.8 | 98.5±12.8 | <0.001 |
| Average Minimal Rim Width, *μm* | 220.5±42.6 | 246.6±47.2 | <0.001 |
| Mean deviation, *dB* | -4.20±3.12 | -1.06±1.48 | <0.001 |
| Pattern standard deviation, *dB* | 5.45±3.68 | 1.97±0.84 | <0.001 |

IOP = intraocular pressure; BMO = Bruch's membrane opening; PPA = parapapillary atrophy; RNFL = retinal nerve fiber layer.

*Comparison performed using paired *t*-test.

difference of offset index and β-zone PPA area, those parameters of the fellow control eyes were subtracted from those of the NTG eyes (Fig 2B). In contrast to the β-zone PPA area, the offset index was consistently larger in the NTG eyes than in the fellow control eyes over the entire range of axial length (Fig 2B).

The angular deviation of the CRVT showed a correlation with the angular location of RNFL defect in the total group ($r = -0.444$, $P<0.001$) and in the subgroup with the temporal location of CRVT ($r = -0.749$, $P = 0.020$).

## Discussion

Newborns have their CRVTs in the central area of the ONH in most cases [13]. In the Boramae Myopia Cohort Study, we demonstrated actual shifting of CRVT in contrast to the preserved BMO during myopic axial elongation [5–7]. Because the CRVT is embedded in the dense connective tissue of the LC [14], shift and deviation of the CRVT from the BMO center would be related to shift and subsequent offset of the underlying LC. Moreover, CRVT deviation from the BMO center is closely related to the location of optic nerve damage in myopic open-angle

**Table 2. Factors associated with presence of glaucoma within each subject.**

|  | Univariable analysis | | | Multivariable analysis* | | |
|---|---|---|---|---|---|---|
|  | OR | 95% CI | P | OR | 95% CI | P |
| Baseline IOP, *mmHg* | 1.067 | (0.946, 1.203) | 0.292 |  |  |  |
| Axial length, *mm* | 1.060 | (0.878, 1.280) | 0.545 |  |  |  |
| Angular deviation of vascular trunk, ° | 1.005 | (0.999, 1.011) | 0.119 | 1.006 | (0.999, 1.012) | 0.084 |
| Offset index | **17.611** | **(4.972, 62.380)** | **<0.001** | **31.625** | **(4.794, 208.619)** | **<0.001** |
| BMO area, $mm^2$ | 1.190 | (0.783, 1.809) | 0.416 |  |  |  |
| β-zone PPA area, $mm^2$ | 1.274 | (0.896, 1.812) | 0.178 | 1.247 | (0.733, 2.123) | 0.415 |
| γ-zone PPA area, $mm^2$ | 1.584 | (0.949, 2.643) | 0.078 | 0.425 | (0.164, 1.105) | 0.079 |

OR = odds ratio; CI = confidence interval; IOP = intraocular pressure; BMO = Bruch's membrane opening; PPA = parapapillary atrophy.

Statistically significant values ($P<0.05$) are shown in bold.

*Variables with $P<0.20$ in the univariate analysis were included in the subsequent multivariate analysis.

Paired eye correlation was adjusted using the generalized linear mixed-effects model.

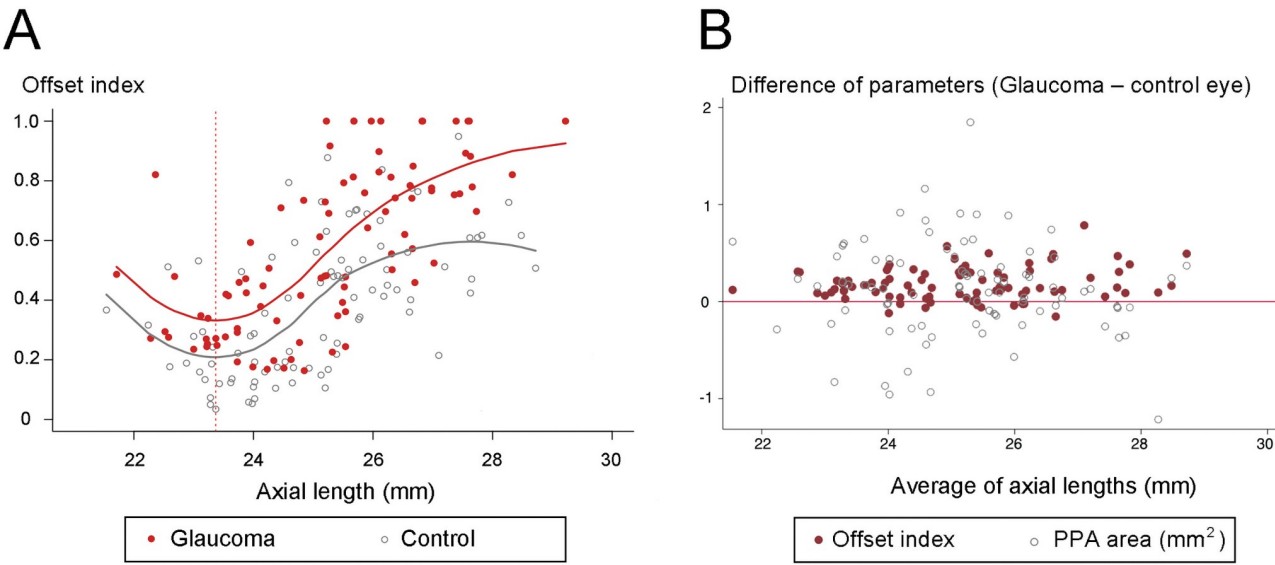

**Fig 2. Offset index according to glaucoma diagnosis and axial length.** (**A**) Scatter plot showing offset index according to diagnosis and axial length. Locally weighted scatterplot smoothing (LOESS) curves were fitted to describe the change of offset index depending on the axial length in each diagnosis. A generalized estimating equation (GEE) regression model revealed that the offset index was larger in the glaucoma eyes while it showed non-linear relationship with axial length (Table 3). The minimum value of offset index was anticipated at an axial length of 23.4 mm (red dashed vertical line). (**B**) Intra-individual difference of offset index and β-zone parapapillary atrophy (PPA) area. The difference was defined as the subtraction of values in the fellow control eyes from those in the glaucoma eyes. Along the reference line (red line), the upper side represents the larger value in glaucoma eyes, while the lower side the lesser value in glaucoma eyes. Compared with the differences of β-zone PPA areas, those of the offset index were located mostly on the upper side.

glaucoma patients [8, 9]. This could be an explanation for the close association between CRVT and the location of glaucomatous damage, which were reported in many other reports [15–18]. Taken together, LC/BMO offset on the en-face plane, possibly as the reason of CRVT deviation from the BMO center, could reflect stress exerted on the LC. In the present study, we found that the CRVT position was deviated farther from the BMO center in glaucoma eyes than in fellow control eyes in unilateral NTG patients. This implies that a larger LC/BMO offset might be more susceptible to glaucomatous damage than a smaller LC/BMO offset within the same subject.

**Table 3. Offset Index according to diagnosis and axial length.**

| | Generalized estimating equation regression model | | |
|---|---|---|---|
| | **Coefficient** | **95% CI** | ***P*** |
| Intercept | **148.756** | **(67.582, 229.929)** | **<0.001** |
| Diagnosis | **0.158** | **(0.119, 0.196)** | **<0.001** |
| Axial length | **-17.702** | **(-27.378, -8.205)** | **<0.001** |
| Axial length × Axial length | **0.698** | **(0.314, 1.081)** | **<0.001** |
| Axial length × Axial length × Axial length | **-0.009** | **(-0.014, -0.004)** | **<0.001** |

CI = confidence interval.

Statistically significant values ($P < 0.05$) are shown in bold.

The final equation is as follows: *Offset Index* = 148.156 + 0.158 × *Diagnosis* − 17.702 × *Axial length* + 0.698 × *Axial length*$^2$ − 0.009 × *Axial length*$^3$.

It implies that the glaucoma eye has a larger offset index than the fellow control eye in a given axial length, while the change of offset index according to axial length is not linear. Our equation estimates the minimum value of offset index in an eye with an axial length of 23.4 mm, which corresponds to the axial length of a near-emmetropic eye.

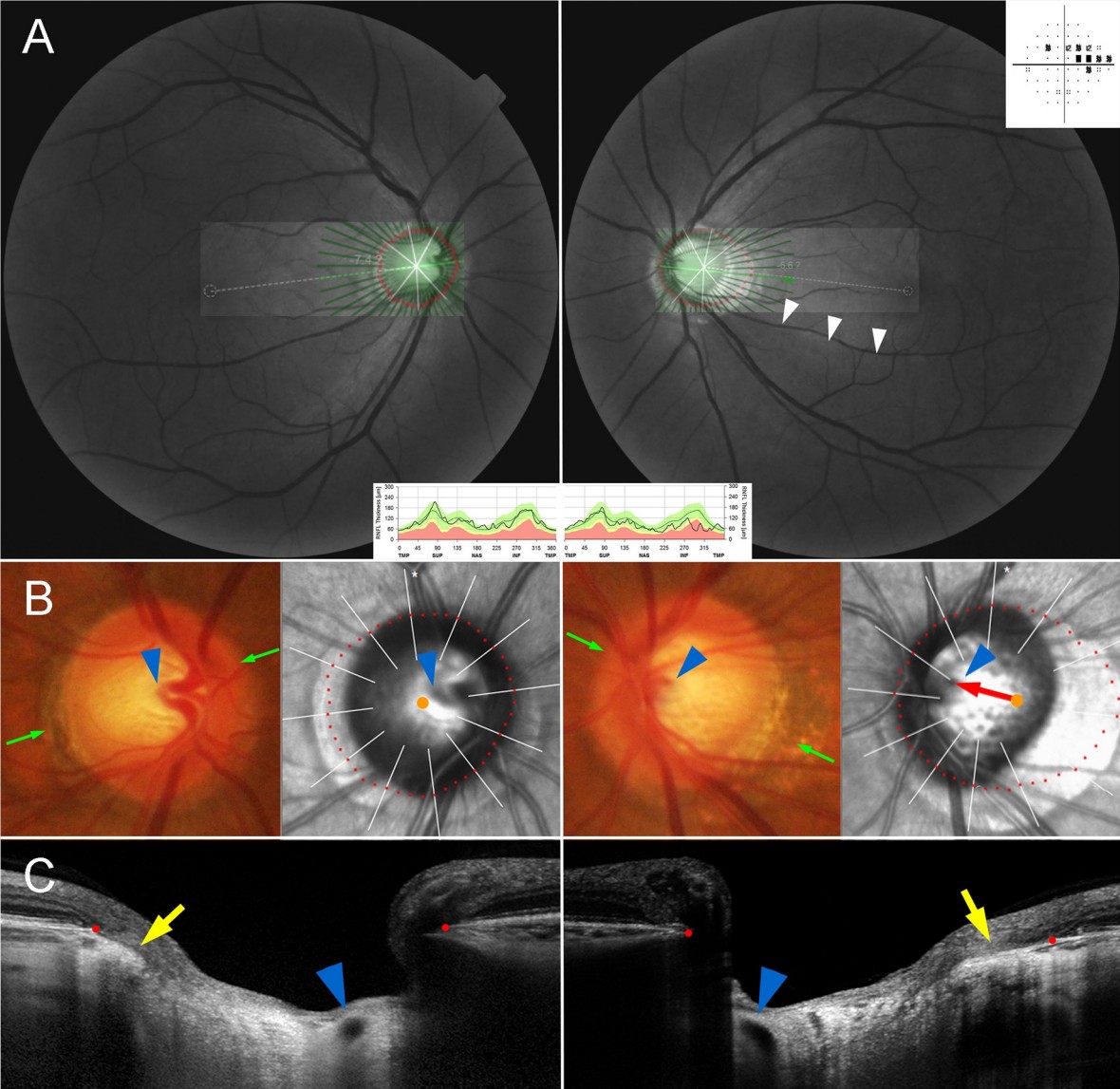

**Fig 3. Inter-eye comparison of central retinal vascular trunk (CRVT) positions located on nasal side.** (**A**) Red-free fundus photos and Humphrey visual field results. The infrared funduscopic spectral-domain (SD) OCT image is transposed to show the Bruch's membrane opening (BMO) margin (red dots). The red-free fundus photos show inferior retinal nerve fiber layer (RNFL) defect in the left eye only (white arrowheads). The Humphrey visual field results show superior scotomas corresponding to the RNFL defect. Circumpapillary RNFL thickness maps, as measured by SD-OCT, are given at the bottom for comparison. (**B**) Disc photographs and infrared fundus images. The red dots indicate the BMO margin. The orange dots indicate the centers of the BMO. The arrowheads indicate the CRVT. The green arrows indicate the locations of the SD-OCT scans. Please note the larger offset in the glaucoma eye (red arrow). (**C**) B-scan SD-OCT images show emergence of CRVTs (arrowheads). Please note the larger offset in the glaucoma eye as evidenced by the position of the CRVT (arrowheads) and the difference of the externally oblique borders (yellow arrows) on the temporal side.

To evaluate the effect of LC/BMO offset on glaucoma development, we included exclusively unilateral NTG patients having one glaucoma eye and one non-glaucoma eye. In this manner, we could exclude the effects of systemic factors such as aging and other general health-related conditions [19]. Subsequently, we focused on the local factors that make eyes more susceptible to glaucomatous damage. Although, the fellow eyes of unilateral NTG patients might differ from the healthy eyes, and unilaterality might exist only within a limited

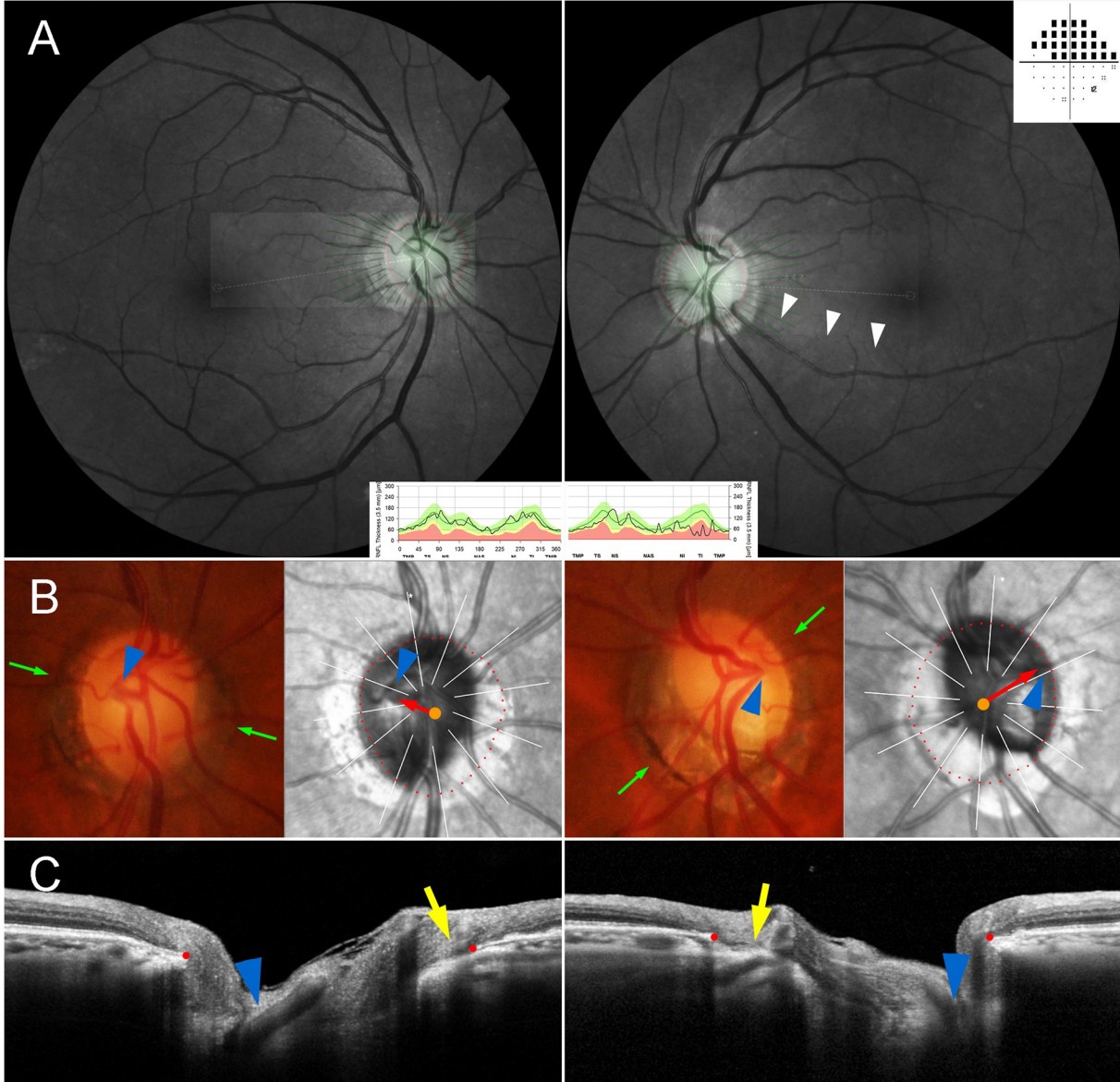

**Fig 4. Inter-eye comparison of central retinal vascular trunk (CRVT) positions located on temporal side.** (**A**) Red-free fundus photos and Humphrey visual field results. The infrared funduscopic spectral-domain (SD) OCT image is transposed to show the Bruch's membrane opening (BMO) margin (red dots). The red-free fundus photos show inferior retinal nerve fiber layer (RNFL) defect in the left eye only (white arrowheads). The Humphrey visual field results show superior scotomas corresponding to the RNFL defect. Circumpapillary RNFL thickness maps, as measured by SD-OCT, are given at the bottom for comparison. (**B**) Disc photographs and infrared fundus images. The red dots indicate the BMO margin. The orange dots indicate the centers of the BMO. The arrowheads indicate the CRVT. The green arrows indicate the locations of the SD-OCT scans. Please note the larger offset in the glaucoma eye (red arrows). (**C**) B-scan SD-OCT images show emergence of CRVTs (arrowheads). Please note the larger offset in the glaucoma eye as evidenced by the position of the CRVT (arrowheads) and the difference of the externally oblique borders (yellow arrows) on the nasal side.

period of time [20], our study, at the time of its conduct, informed us which eye was more susceptible to damage: the eye with larger LC/BMO offset was affected by glaucomatous damage earlier than the fellow eye with smaller LC/BMO offset. This might explain why glaucoma occurs in the eye that it does, only under the same systemic risk factors and even within the normal range of IOP.

LC/BMO offset can increase the susceptibility of the ONH to glaucomatous damage, because it reflects the tangential stress that had been applied to the ONH during eyeball growth [5–9]. Interestingly, in the present study, LC/BMO offset did not show a linear correlation with axial length (Fig 2A). Rather, the correlation between LC/BMO offset and axial length was a J-shaped curve, and the GEE regression model showed that the offset index was the smallest for the axial length of 23.4 mm (Table 3), from which point, it increases in either direction. This could be understood by the various shapes of ocular expansion during growth [10]. The average axial length of newborns is around 17 mm, which is about 7 mm shorter than that of adults [21]. This means that every eyeball has to grow after birth, regardless of how short it will be in adulthood. Therefore, the process of outer-wall shift would not be limited to myopic eyes only, but would manifest in eyes of larger-than-17 mm axial length as well. In contrast to the prolate (= axial overgrowth) growth of myopic eyes [10, 22], some eyes, especially hyperopic ones, have been reported to have oblate growth [10, 23]. As we speculated in our previous study [10], oblate expansion of the eyeball would lead to temporal shift of the outer-wall (including LC) relative to the posterior polar retinal structure (including BMO) if the latter is selectively preserved during the expansion. We speculated that temporal shift of the outer-wall, as related to oblate growth in hyperopic eyes, could be the cause of offset index increase in the reverse direction in eyes of axial length less than 23.4 mm (Fig 2A).

Two of our previous studies explored the association of glaucomatous damage and offset direction in myopic eyes [8, 9]. The offset, however, was not confined to myopia but existed even in hyperopic eyes, as stated above. Our current study showed that such offset, across the entire range of axial length, was larger in the NTG group than in the control group (Table 3). The GEE regression model revealed that the offset index of the NTG eye was larger than that of the control eye for a given axial length (Fig 2A). This suggested that the NTG eyes had to endure more shifting than the control eyes with similar axial lengths. To summarize, LC/BMO offset could be an indicator of cumulative tensile stress exerted on the ONH during growth, and thereby might increase the susceptibility to glaucoma not in myopic eyes only but in any eyes.

It should be noted that the direction of LC/BMO offset was associated with the location of glaucomatous damage in cases either of nasal or temporal direction offsets. The large pores of the LC in the superior and inferior regions are considered to be more susceptible to glaucomatous damage [4, 14, 24]. Moreover, the LC is reported to have a horizontal ridge [14, 25], which might protect the LC from tensile stress acting parallel to it. Therefore, we speculated that tensile stress would converge to the susceptible pores in the superior and inferior regions [8, 9], even if the direction of LC/BMO offset was nearly parallel to the horizontal meridian.

Park et al. showed by means of a subgroup analysis that β-zone-PPA-associated variables could be risk factors for unilateral NTG [26]. Also in our study, the β-zone PPA area was larger in the NTG group, though it was not significant in the multivariable analysis. In our previous study, we showed that a part of β-zone PPA, which is to say, not only γ-zone PPA but also some of β-zone PPA with Bruch's membrane, appeared as the manifestation of LC and scleral shifting beneath the preserved BMO [7]. In this type of β-zone PPA, the extent of LC offset, as measured by CRVT dragging, was larger than the extent of β-zone PPA change [7]. When restricted to γ-zone PPA within the β-zone, the association with glaucoma diagnosis was only marginally significant. We speculated that, in the case of either β-zone or γ-zone PPA, the effect of PPA might have been smeared out by the larger amount of CRVT deviation, which is more representative of LC/BMO offset (Fig 2B) [7]. The following implication of β-zone PPA, however, should be noted: at least in some eyes, larger β-zone PPA would represent a larger LC offset below the BMO, which makes the ONH more susceptible to a second insult such as increased IOP or other tissue-toxic factors [1, 3].

NTG has several distinctive characteristics. The ONH of NTG is affected by glaucoma even within the normal range of IOP [27]. RNFL defect has a tendency to be localized, in contrast to the diffuse atrophy of high-tension glaucoma [28–30]. A hemispheric location of glaucomatous damage has been reported to be asymmetric in NTG eyes, while high-tension glaucoma incurred similar damage in both hemispheres [31, 32]. Furthermore, reduction of IOP remains effective as a treatment of NTG [33]. The concept of LC/BMO offset could explain those characteristics of NTG. First, LC/BMO offset would increase the locoregional susceptibility of the given ONH, as noted above. Second, localized RNFL defect and asymmetry of the hemispheric location of damage could be understood by the directionality of offset: vulnerable pores in the hemisphere of the opposite direction of shifting would be selectively damaged in NTG eyes. Finally, IOP reduction could be effective, since it can reduce both the direct IOP-related axial force and the tensile stress exerted on the outer load-bearing structures, which might act as a tangential force exacerbating, by the shearing effect, damage to the ONH.

This study has several limitations. First, the study design was cross-sectional. As such, we could not demonstrate actual LC shifting during earlier growth periods. The premise of this study was based on our previous prospective cohort study results [5–7]. Thus, we cannot exclude the possibility of confounding effect of LC remodeling on the CRVT position in the NTG eyes. However, glaucomatous ONH change has been reported not to affected the CRVT position in the LC portion [9, 34]. Second, the location of the CRVT within the BMO, which was used as the indicator of LC/BMO offset, has a limitation. Since the CRVT outside the BMO could not be visualized with angiography, the extent of LC/BMO offset could be underestimated in such cases. Moreover, some eyes showed bifurcation of the CRVT on its emergence. Since we excluded such cases, we did not have any means of measuring the LC/BMO offset in those eyes. Additionally, the initial location of the CRVT was presumed to be the BMO center, which would not be certain for all eyes. However, most newborns had the CRVT in the center of the optic disc [13], and the hyaloid artery is reported to be in the middle of the orbital part of the nerve when the back of the globe is formed in the embryo state [35]. Third, we did not consider the three-dimensional eyeball shape in evaluating LC/BMO offset. Recent studies have brought attention to posterior scleral shape in association with optic disc morphology and glaucoma development [36–38]. Although, we considered, as stated in our previous report [10], that LC/BMO offset might represent the three-dimensional eyeball shape, three-dimensional eyeball shape and neural canal obliqueness [39, 40] (as opposed to two-dimensional offset) is a topic better left to a future study. Finally, we exclusively included subjects with unilateral NTG. Unilateral and bilateral NTG patients might have different characteristics [19]. Patients with systemic risk factors might have bilateral glaucomatous damage [19], though they were selectively excluded from this study. Therefore, our relevant study results should be interpreted with caution.

In conclusion, the LC had been shifted farther from the BMO center in the glaucoma eye than in the fellow control eye of unilateral NTG patients. Larger LC/BMO offset might represent the larger cumulative stress exerted on the ONH during growth, which makes the ONH more susceptible to additional insults induced by increased IOP or other factors.

## Author Contributions

**Conceptualization:** Martha Kim, Kyoung Min Lee, Seok Hwan Kim.

**Data curation:** Sohee Oh, Kyoung Min Lee, Seok Hwan Kim.

**Formal analysis:** Sohee Oh, Kyoung Min Lee.

**Funding acquisition:** Kyoung Min Lee.

**Investigation:** Ho-Kyung Choung, Kyoung Min Lee, Seok Hwan Kim.

**Methodology:** Kyoung Min Lee, Seok Hwan Kim.

**Resources:** Seok Hwan Kim.

**Supervision:** Ho-Kyung Choung, Kyoung Min Lee, Seok Hwan Kim.

**Visualization:** Kyoung Min Lee.

**Writing – original draft:** Ho-Kyung Choung, Kyoung Min Lee, Seok Hwan Kim.

**Writing – review & editing:** Ho-Kyung Choung, Martha Kim, Kyoung Min Lee, Seok Hwan Kim.

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
