## [Decision Letter · Decision Letter 0]

14 Jun 2021

PONE-D-21-07989

Central retinal vascular trunk deviation in unilateral normal-tension glaucoma

PLOS ONE

Dear Dr. Lee,

Thank you for submitting your manuscript to PLOS ONE. After careful consideration, we feel that it has merit but does not fully meet PLOS ONE’s publication criteria as it currently stands. Therefore, we invite you to submit a revised version of the manuscript that addresses the points raised during the review process.

I would like to draw your attention to comments from Reviewer #2 which I would like to see addressed in your revision. 

We look forward to receiving your revised manuscript.

Kind regards,

Simon J Clark, D.Phil.

Academic Editor

PLOS ONE

Journal Requirements:

2.Thank you for stating the following in your Competing Interests section: 

"No"

Reviewers' comments:

Reviewer's Responses to Questions

**Comments to the Author**

1. Is the manuscript technically sound, and do the data support the conclusions?

Reviewer #1: Yes

Reviewer #2: Partly

2. Has the statistical analysis been performed appropriately and rigorously? 

Reviewer #1: Yes

Reviewer #2: N/A

3. Have the authors made all data underlying the findings in their manuscript fully available?

Reviewer #1: Yes

Reviewer #2: Yes

4. Is the manuscript presented in an intelligible fashion and written in standard English?

Reviewer #1: Yes

Reviewer #2: Yes

5. Review Comments to the Author

Reviewer #1: The authors investigated whether the position of the CRVT increases susceptibility to glaucomatous damage in unilateral NTG. The LC shifted farther from BMO center in the glaucoma eye than in the fellow control eye. The manuscript was well written. Therefore, I have no comments.

Reviewer #2: 1. Authors showed that position of the central retinal vascular trunk (CRVT) had good correlation with the presence of glaucoma in normal tension glaucoma (NTG) patients. And calculation of CRVT, measured as the deviation from the center of the Bruch’s membrane opening (BMO) was well understood. But even considering, the relative preservation of the Bruch’s membrane, since the BMO, which is the reference value, is not an absolute representative value, it is questionable whether it is possible to assume the CRVT dependent on it as an independent value.

2. I wonder if it is okay to understand CRVT (surrogate of LC shift) as an intra optic disc factor. If so, was it compared with other factors such as the posterior pole index in addition to the PPA beta zone, which is the peri optic disc factor.

3. In Table 1, a monocular NTG patient with myopia was targeted in this study to compare NTG eye and normal eye. It seems that there is no significant axial length difference between the eyes. If so, isn't the BMO preservation mentioned in line 47 a contrary, which should be preserved better compared to the shift index? Additionally, there is only the VF results compared between the two groups, and I wonder if the degree of RNFL damage was further analyzed.

4. The deepest point seems to be different for each eyeball, according to your study. (Relationship between Three-Dimensional Magnetic Resonance Imaging Eyeball Shape and Optic Nerve Head Morphology) So I wonder how the correction was made when measuring BMO and CRVT, which are observational values on a plane.

5. In line 323, you mentioned the analysis of the beta zone PPA. I wonder if the smear-out result was the same when compared to the CRVT and other zones of PPA other than beta zone.

6. Lastly, if you look at the previous studies you conducted, overlapping information regarding CRVT is also shown in this study. Therefore, I would like you to describe in detail the contents of this study that differentiate it from previous studies.

6. PLOS authors have the option to publish the peer review history of their article (what does this mean?). If published, this will include your full peer review and any attached files.

Reviewer #1: No

Reviewer #2: No

---

## [Author Response · Author response to Decision Letter 0]

1 Jul 2021

Editor comments:

Thank you for submitting your manuscript to PLOS ONE. After careful consideration, we feel that it has merit but does not fully meet PLOS ONE’s publication criteria as it currently stands. Therefore, we invite you to submit a revised version of the manuscript that addresses the points raised during the review process. I would like to draw your attention to comments from Reviewer #2 which I would like to see addressed in your revision. 

Above all, thank you very much for giving us an opportunity to revise this manuscript. Every issue will be dealt with in turn and in detail. 

We have checked it. 

2.Thank you for stating the following in your Competing Interests section: 

"No"

O.K. 

O.K.

 

Reviewer #1: 

The authors investigated whether the position of the CRVT increases susceptibility to glaucomatous damage in unilateral NTG. The LC shifted farther from BMO center in the glaucoma eye than in the fellow control eye. The manuscript was well written. Therefore, I have no comments.

Thank you very much for your kind comments. We are really appreciative.

 

Reviewer #2: 

1. Authors showed that position of the central retinal vascular trunk (CRVT) had good correlation with the presence of glaucoma in normal tension glaucoma (NTG) patients. And calculation of CRVT, measured as the deviation from the center of the Bruch’s membrane opening (BMO) was well understood. But even considering, the relative preservation of the Bruch’s membrane, since the BMO, which is the reference value, is not an absolute representative value, it is questionable whether it is possible to assume the CRVT dependent on it as an independent value. 

First of all, we deeply appreciate your brilliant and incisive comment, which has improved our manuscript. This is a really important point. As the reviewer pointed out, the preservation of the BMO was only relative to the outer scleral layer, and the BMO is not an absolute reference point. In fact, we cannot set an absolute reference point to determine optic nerve head (ONH) change during growth in any way, since every change has relativity: the CRVT is shifted from the point of the BMO center, while the BMO is shifted from the point of the CRVT. What we wanted to represent was misalignment between the BMO and LC, which was larger in the glaucoma eyes. To clarify this, we have changed “the LC shift” to “the LC/BMO offset” and “shift index” to “offset index” throughout the entire manuscript to emphasize that what is important is the misalignment between the LC and BMO, not the individual absolute location. Thank you for your indispensable comment. 

2. I wonder if it is okay to understand CRVT (surrogate of LC shift) as an intra optic disc factor. If so, was it compared with other factors such as the posterior pole index in addition to the PPA beta zone, which is the peri optic disc factor.

As stated above, we do not think that LC/BMO offset is a factor restricted to the optic disc; rather, the offset is determined by alignment change between the retinal and scleral layers in the ONH. In our previous study (Ref #10), we showed that such alignment changes could be induced by three-dimensional expansion of the eyeball during growth periods. Therefore, LC/BMO offset might not be an intra-optic disc factor, but rather, it alone may represent pan-globe changes (including peri optic disc factors as well as posterior polar changes) during eyeball expansion. However, we think that the point addressed by the reviewer is worth studying in the future. Therefore, we addressed it as a limitation as follows (page 19 lines 367–372):

“Third, we did not consider the three-dimensional eyeball shape in evaluating LC/BMO offset. Recent studies have brought attention to posterior scleral shape in association with optic disc morphology and glaucoma development. Although we considered, as stated in our previous report, that LC/BMO offset might represent the three-dimensional eyeball shape, three-dimensional eyeball shape and neural canal obliqueness (as opposed to two-dimensional offset) is a topic better left to a future study.”

3. In Table 1, a monocular NTG patient with myopia was targeted in this study to compare NTG eye and normal eye. It seems that there is no significant axial length difference between the eyes. If so, isn't the BMO preservation mentioned in line 47 a contrary, which should be preserved better compared to the shift index? Additionally, there is only the VF results compared between the two groups, and I wonder if the degree of RNFL damage was further analyzed.

First, as demonstrated in the scatter plot (Fig. 2), the subject of this study was not restricted to myopia. Second, Table 1 presented the intra-individual comparisons between glaucoma and fellow control eyes. Therefore, it can say nothing about the degree of BMO preservation against axial length increase. It concerns only whether BMO area was larger or axial length was longer in glaucoma eyes than in control eyes. 

Prompted by the reviewer’s suggestion, we have constructed generalized estimating equation modeling to evaluate the effects of axial length, diagnosis, and their interaction on the BMO area, while adjusting the paired-eye correlations. 

BMO area, mm2 Coefficient 95% CI P

Axial length, mm 0.168 (0.085, 0.250) <0.001

Diagnosis -0.607 (-1.832, 0.618) 0.331

Axial length × Diagnosis 0.026 (-0.022, 0.075) 0.291

BMO area was associated with axial length. Neither glaucoma diagnosis nor the interaction between axial length and diagnosis was associated with BMO area. This means that BMO area increases with axial length increase, but that the slopes of BMO area change according to the axial length increase were not different between the glaucoma and control groups. It is well known that BMO diameter increases rapidly with axial length increase over 26mm [1]. “BMO preservation” means that there is less change of it relative to the LC and scleral tissue during eyeball expansion. In the eyes of axial length less than 26mm, the average BMO area was 2.61±0.72 for the glaucoma group, and 2.61±0.70 for the control group. In the eyes of axial length greater than 26mm, the average BMO area was 3.06±0.63 for the glaucoma group, and 2.88±0.55 for the control group. To summarize, the BMO area was larger in high myopia, but the effect of high myopia did not differ between glaucoma and control groups.

 Zhang Q, Xu L, Wei WB, Wang YX, Jonas JB. Size and Shape of Bruch's Membrane Opening in Relationship to Axial Length, Gamma Zone, and Macular Bruch's Membrane Defects. Investigative ophthalmology & visual science. 2019;60(7):2591-8. Epub 2019/06/21. doi: 10.1167/iovs.19-27331. PubMed PMID: 31219533.

The degree of axonal damage other than visual field results was another issue raised by this question. Inspired your brilliant suggestion, we have added information on not only circumpapillary RNFL thickness but also BMO-MRW thickness in the Methods section (page 7 lines 160–163):

“Based on deep ONH scanning using the Glaucoma Module Premium Edition of the Spectralis machine, the BMO minimum rim width (BMO-MRW) and circumpapillary RNFL thickness of 3.5mm diameter from the BMO center were also measured.”

The results of this comparison are now included in the Results section (page 9 lines 191–192, Page 10 Table 1):

“thinner average circumpapillary RNFL thickness (85.2 ± 13.8 µm vs. 98.5 ± 12.8 µm, P<0.001), narrower average BMO-MRW (220.5 ± 42.6 µm vs. 246.6 ± 47.2 µm, P<0.001)”

Thank you very much for your excellent comments.

4. The deepest point seems to be different for each eyeball, according to your study. (Relationship between Three-Dimensional Magnetic Resonance Imaging Eyeball Shape and Optic Nerve Head Morphology) So I wonder how the correction was made when measuring BMO and CRVT, which are observational values on a plane.

This point is also related to questions #1 and #2. Please refer to our answers above. In our previous study (Ref #10), we showed that the three-dimensional eyeball shape, which includes spherical shapes (prolate, horizontally oblate, and vertically oblate) and asymmetric protrusion of the eyeball, determines the LC/BMO offset direction. Therefore, we think that LC/BMO offset itself might be understood as a two-dimensional translation of the three-dimensional eyeball shape. Regarding the offset, Dr. Burgoyne’s group also has evaluated the alignment between the anterior scleral opening (ASCO) and BMO [1, 2]. Although they evaluated the neural canal obliqueness three-dimensionally, the offset between the ASCO and the BMO was also defined on the two-dimensional plane with projection. Since we think that LC/BMO offset is determined by the three-dimensional eyeball shape, adjusting the three-dimensional eyeball shape again might result in over-adjustment. 

 Hong S, Yang H, Gardiner SK, Luo H, Hardin C, Sharpe GP, et al. OCT-Detected Optic Nerve Head Neural Canal Direction, Obliqueness, and Minimum Cross-Sectional Area in Healthy Eyes. American journal of ophthalmology. 2019;208:185-205. Epub 2019/05/17. doi: 10.1016/j.ajo.2019.05.009. PubMed PMID: 31095953; PubMed Central PMCID: PMCPMC6851461.

 Jeoung JW, Yang H, Gardiner S, Wang YX, Hong S, Fortune B, et al. Optical Coherence Tomography Optic Nerve Head Morphology in Myopia I: Implications of Anterior Scleral Canal Opening Versus Bruch Membrane Opening Offset. American journal of ophthalmology. 2020;218:105-19. doi: https://doi.org/10.1016/j.ajo.2020.05.015.

Nonetheless, we think that the issue raised by the reviewer is quite reasonable. Therefore, we clarified this issue in the limitations section (page 19 lines 367–372), as stated above, with proper reference citation for those who are not familiar with the deepest point of the eye [1–3]. 

1. Kim YC, Jung Y, Park HL, Park CK. The Location of the Deepest Point of the Eyeball Determines the Optic Disc Configuration. Sci Rep. 2017;7(1):5881. Epub 2017/07/21. doi: 10.1038/s41598-017-06072-8. PubMed PMID: 28725046; PubMed Central PMCID: PMCPMC5517507.

2. Kim YC, Koo YH, Bin Hwang H, Kang KD. The Shape of Posterior Sclera as a Biometric Signature in Open-angle Glaucoma: An Intereye Comparison Study. J Glaucoma. 2020;29(10):890-8. Epub 2020/06/20. doi: 10.1097/ijg.0000000000001573. PubMed PMID: 32555059; PubMed Central PMCID: PMCPMC7647446.

3. Jeon SJ, Park HL, Kim YC, Kim EK, Park CK. Association of Scleral Deformation Around the Optic Nerve Head With Central Visual Function in Normal-Tension Glaucoma and Myopia. Am J Ophthalmol. 2020;217:287-96. Epub 2020/05/11. doi: 10.1016/j.ajo.2020.04.041. PubMed PMID: 32387433.

5. In line 323, you mentioned the analysis of the beta zone PPA. I wonder if the smear-out result was the same when compared to the CRVT and other zones of PPA other than beta zone.

That is a really good qeustion. Accordingly, we have revised the definition of PPA as follows (page 7 lines 154–157; lines 159–160):

“β-zone parapapillary atrophy (PPA) was defined as the area without any retinal pigment epithelium (RPE) adjacent to the ONH, and γ-zone PPA as the area without BM within the β-zone PPA. The β-zone and γ-zone PPA areas were measured on the same infrared OCT fundus images by one observer (KML) blinded to the subjects’ information.”

“The β-zone PPA area was defined as the RPEO area minus the CDM area, while the γ-zone PPA was defined as the BMO area minus the CDM area.”

The analyses incorporating the γ-zone PPA area have been added to the Results section (page 9 line 196; lines 199–200; page 10 Table 1; page 11 Table 2):

“and that of γ-zone PPA (0.62 ± 0.62 mm2 vs. 0.46 ± 0.54 mm2, P<0.001)”

“The GLMM analysis revealed that larger offset index was the only risk factor for NTG diagnosis (OR = 31.625, P<0.001) (Table 2).”

The meaning of such results has been explained in the Discussion (page 18 lines 335–339).

“In this type of β-zone PPA, the extent of LC offset, as measured by CRVT dragging, was larger than the extent of β-zone PPA change. When restricted to γ-zone PPA within the β-zone, the association with glaucoma diagnosis was only marginally significant. We speculated that, in the case of either β-zone or γ-zone PPA, the effect of PPA might have been smeared out by the larger amount of CRVT deviation, which is more representative of LC/BMO offset (Fig. 2B).”

Thank you for your invaluable comments.

6. Lastly, if you look at the previous studies you conducted, overlapping information regarding CRVT is also shown in this study. Therefore, I would like you to describe in detail the contents of this study that differentiate it from previous studies.

We understand your confusion, since the main idea running through the current and previous studies is similar. The originality of this study was that the subjects were not confined to myopia cases, as stated in the Discussion (page 17 lines 319–322). The average axial length of newborns is around 17mm, which is about 7mm shorter than that of adults (Ref #21). Therefore, expansion of the eyeball is not limited to myopia; rather, it occurs in all eyes, including hyperopic eyes. Our previous 3D-MRI study revealed an association of offset direction and eyeball shape for all ranges of axial length (Ref #10). In this study, we showed that the offset, which occurs in all eyes, might increase susceptibility to glaucoma when other systemic factors are adjusted, as they were for our unilateral NTG eyes (page 16 lines 287–296). To clarify this, we have revised one paragraph to explain the originality of this study in more detail (page 17 lines 314–322): 

“Two of our previous studies explored the association of glaucomatous damage and offset direction in myopic eyes. The offset, however, was not confined to myopia but existed even in hyperopic eyes, as stated above. Our current study showed that such offset, across the entire range of axial length, was larger in the NTG group than in the control group (Table 3). The GEE regression model revealed that the offset index of the NTG eye was larger than that of the control eye for a given axial length (Fig. 2A). This suggested that the NTG eyes had to endure more shifting than the control eyes with similar axial lengths. To summarize, LC/BMO offset could be an indicator of cumulative tensile stress exerted on the ONH during growth, and thereby might increase the susceptibility to glaucoma not in myopic eyes only but in any eyes.”

Thank you very much for your incisive comments.

---

## [Decision Letter · Decision Letter 1]

7 Jul 2021

Central retinal vascular trunk deviation in unilateral normal-tension glaucoma

PONE-D-21-07989R1

Dear Dr. Lee,

We’re pleased to inform you that your manuscript has been judged scientifically suitable for publication and will be formally accepted for publication once it meets all outstanding technical requirements.

Kind regards,

Simon J Clark, D.Phil.

Academic Editor

PLOS ONE

Additional Editor Comments (optional):

Reviewers' comments:

Reviewer's Responses to Questions

**Comments to the Author**

1. If the authors have adequately addressed your comments raised in a previous round of review and you feel that this manuscript is now acceptable for publication, you may indicate that here to bypass the “Comments to the Author” section, enter your conflict of interest statement in the “Confidential to Editor” section, and submit your "Accept" recommendation.

Reviewer #1: (No Response)

Reviewer #2: All comments have been addressed

2. Is the manuscript technically sound, and do the data support the conclusions?

Reviewer #1: (No Response)

Reviewer #2: Yes

3. Has the statistical analysis been performed appropriately and rigorously? 

Reviewer #1: (No Response)

Reviewer #2: Yes

4. Have the authors made all data underlying the findings in their manuscript fully available?

Reviewer #1: (No Response)

Reviewer #2: Yes

5. Is the manuscript presented in an intelligible fashion and written in standard English?

Reviewer #1: (No Response)

Reviewer #2: Yes

6. Review Comments to the Author

Reviewer #1: (No Response)

Reviewer #2: I think that the additional explanation in the thesis can be a good information delivery to the readers. Thank you for your hard work.

7. PLOS authors have the option to publish the peer review history of their article (what does this mean?). If published, this will include your full peer review and any attached files.

Reviewer #1: No

Reviewer #2: No

---

## [Editor Report · Acceptance letter]

9 Jul 2021

PONE-D-21-07989R1 

Central retinal vascular trunk deviation in unilateral normal-tension glaucoma 

Dear Dr. Lee:

I'm pleased to inform you that your manuscript has been deemed suitable for publication in PLOS ONE. Congratulations! Your manuscript is now with our production department. 

Kind regards, 

on behalf of

Prof. Simon J Clark 

Academic Editor

PLOS ONE